# Electrosynthesized Poly(*o*-aminophenol) Films as Biomimetic Coatings for Dopamine Detection on Pt Substrates

**Rosanna Ciriello \*, Martina Graziano, Giuliana Bianco**  **and Antonio Guerrieri** 

Dipartimento di Scienze, Università degli Studi della Basilicata, Viale dell'Ateneo Lucano 10, 85100 Potenza, Italy; martina.graziano@studenti.unibas.it (M.G.); giuliana.bianco@unibas.it (G.B.); antonio.guerrieri@unibas.it (A.G.)
\* Correspondence: rosanna.ciriello@unibas.it

**Abstract:** Dopamine (DA) is a neurotransmitter, and its levels in the human body are associated with serious diseases. The need for a suitable detection method in medical practice has encouraged the development of electrochemical sensors that take advantage of DA electroactivity. Molecularly imprinted polymers (MIPs) are biomimetic materials able to selectively recognize target analytes. A novel MIP sensor for DA is proposed here based on a thin film of poly(*o*-aminophenol) electrosynthesized on bare Pt. A fast and easy method for executing the procedure for MIP deposition has been developed based on mild experimental conditions that are able to prevent electrode fouling from DA oxidation products. The MIP exhibited a limit of detection of 0.65 µM, and appreciable reproducibility and stability. The high recognition capability of poly(*o*-aminophenol) towards DA allowed for the achievement of notable selectivity: ascorbic acid, uric acid, serotonin, and tyramine did not interfere with DA detection, even at higher concentrations. The proposed sensor was successfully applied for DA detection in urine samples, showing good recovery.

**Keywords:** dopamine; electrochemical sensor; molecularly imprinted polymers; poly(*o*-aminophenol); Pt electrode; urine

## 1. Introduction

Dopamine, 3,4-dihydroxyphenyl ethylamine (DA), is an important neurotransmitter that plays significant roles in the central nervous system (CNS), and in the renal, hormonal, and cardiovascular systems [1]. Abnormal DA levels in the human body, either an excess or deficiency, are associated with serious diseases, such as Parkinson's disease (PD), schizophrenia, and drug addiction, and may be indicative of the onset of neuroblastoma and ganglioneuroblastoma [2–6]. In particular, PD is one of the most common age-related neurodegenerative disorders characterized by the progressive degeneration of dopaminergic neurons in the substantia nigra pars compacta and a decrease in striatal dopamine levels. These changes lead to several clinical symptoms: rigidity, resting tremor, and bradykinesia. Although the cause of PD remains unclear, it is widely accepted that, among others, oxidative stress is involved in the pathophysiology of the disease. In recent years, there has been a growing interest in naturally occurring substances able to protect against oxidative stress and dopamine depletion [7]. At the other end, much effort has been devoted to the development of analytical methodologies able to assure an accurate determination of DA, which is of crucial relevance for the diagnosis of the disease.

The techniques available for such detection are mostly based on gas chromatography-mass spectrometry [8]; high-performance liquid chromatography coupled with mass spectrometry [9–11]; fluorescence detection [12]; and chemiluminescence [13]. These techniques allow for extremely low detection limits, even if they pose limitations, such as time consumption, a complicated sample pretreatment, and expensive instrumentation with low portability.

As an alternative, electrochemical methods have drawn notable attention since DA is an electrochemically active molecule that facilitates its detection and they offer advantages

such as ease of operation, low-cost, quick response and feasibility to miniaturization [14]. Electrochemical detection ensures high sensitivity. Nevertheless, it is limited by faradic interference problems from endogenous species electroactive in the potential range adopted for analyte detection. Among them, ascorbic acid and uric acid coexist with DA in biological fluids in 2–3 fold higher concentration levels. Furthermore, the reaction products of DA oxidation are easily adsorbed on the surface of bare electrodes, causing their passivation, which reduces the sensitivity and selectivity of DA detection [15].

To overcome these drawbacks, electrochemical sensors based on molecularly imprinted polymers (MIPs) have been developed. Practically, the surface of conventional electrodes is modified with synthetic biomimetic materials able to selectively recognize the target analyte. These synthetic materials are more stable, robust, and cheaper than natural receptors and offer an equivalent affinity and sensitivity in the molecular recognition of the target analyte. An efficient tool for their realization is electrosynthesis, which allows for the control of the spatial distribution and the thickness of the polymer film by adjusting the electrochemical parameters employed during its deposition.

Nowadays, several MIPs have been proposed for the electrochemical detection of DA [16]. Poly(*o*-aminophenol) (PoAP) is an interesting thin film employed in molecular imprinting technology for the detection of various analytes of clinical and environmental interest. Currently, there are two contributions concerning the involvement of PoAP in the realization of MIPs for DA. In one case, the polymer was electrosynthesized on a conventional gold electrode by cyclic voltammetry at pH 5.5, showing high selectivity towards DA even in the presence of high concentrations of ascorbic acid [17]. More recently, a solid seamless self-supported nanoporous alloy microrod was produced on which PoAP was electrosynthesized as well by cyclic voltammetry at pH 5 [18]. The nanoporous form of the working electrode allowed DA detection at trace levels in biological samples, even if a time-consuming procedure was required for its preparation.

Electrode material is a key factor in determining sensor performances. Almost all reports on MIPs for DA are based on the employment of gold and glassy carbon substrates as the bare electrode or are often combined with nanomaterials to enhance the surface area. As is well known, nanometric dimensions greatly enhance surface reactivity with respect to bulk materials. The unusual properties of these materials are correlated to the nanometer size and can be modulated by varying the dimension within the nanometer scale [19].

The use of platinum as a working electrode is almost unexplored. Indeed, there is only one contribution [20] in which a Pt microbar was modified with a layer of overoxidized poly(pyrrole) to realize a MIP for DA showing high sensitivity and selectivity. As is well known, poly(pyrrole) acquires notable permselective behavior upon electrochemical overoxidation so as to be used for the analysis of complex real samples, such as human sera [21,22].

Platinum certainly plays an important role in electrochemical detection because of its high robustness and reactivity towards DA. However, it is particularly prone to fouling problems caused by the formation of polymeric species in the electrolyte that precipitate on the electrode surface as a result of their large size and high molecular weight. DA itself oxidizes to reaction products that can lead to the formation of melanin-like polymeric structures adhering to the electrode surface [23]. Recently, it has been demonstrated that, at physiological pH, the irreversible adsorption of dopamine on platinum surfaces can occur without necessarily leading to the formation of polymeric materials [24].

Platinum has been widely used in our research group as a support for the electrosynthesis of insulating PoAP. An in-depth study was carried out on the PoAP formation mechanism, which allowed for the hypothesis of a defined chemical structure for the polymer [25]. Knowledge of the chemical composition of polymer chains helps to identify the feasible interactions responsible for the template entrapment inside the polymer cavities, which underlie the imprinting technology.

The aim of the present work was to exploit our previous information on the growth route and composition of PoAP to realize an electrochemical MIP for DA. The novelty

of the proposed sensor lies in the combined use of PoAP as an imprinted polymer and platinum as the electrode material. This has never been explored before.

A preliminary deep investigation into the electrochemical behavior of dopamine on platinum was crucial for the MIP realization. Experimental conditions that were able to prevent electrode fouling from the oxidative products of DA, and that were able to avoid any involvement of the template molecule in the film formation, were evaluated. This last feature is an essential requirement for ensuring successive analyte release from the polymer.

The employment of PoAP as imprinted polymer allowed us to simplify the MIP preparation procedure with respect to the widely explored poly(pyrrole) [20]. Indeed, PoAP in neutral and mild acidic media grows as an insulating polymer, having a compact and pinhole-free structure that hinders the access to the electrode surface of any electroactive compounds that could generate faradic interference. Furthermore, its thin thickness, of around ten nanometers [25], ensures extremely fast response times.

Poly(pyrrole), on the other hand, is electrosynthesized as a conducting polymer and must successively undergo an overoxidation process in order to acquire permselective characteristics. Overoxidation can take place by applying to the polymer an oxidation potential for a prolonged time (at least seven hours) [21,22], or by cycling in a strong alkaline solution, which could affect the template stability. As an example, undesirable DA polymerization is notably hastened in alkaline media, whereas it is negligible in the pH conditions we employed for PoAP electrosynthesis. In this work, we demonstrate that a one-step procedure of about ten minutes ensures the realization of a polymer with built-in permselectivity, and that is ready to use.

Furthermore, it is worth noting that the mechanisms of exclusion controlling the permselective behavior of PoAP and the overoxidized poly(pyrrole) are quite different. An electrostatic control on analyte permeation has been reported in the case of overoxidized poly(pyrrole) [20]. Upon overoxidation, carbonyl and carboxylic functionalities are incorporated into the polymer backbone, which repels anionic molecules, such as ascorbic acid and uric acid, while attracting species with positive groups, such as -NH$_3$ in DA. Unfortunately, structural analogues of DA, such as tyramine and serotonin, have the same positive group as well. In the case of PoAP, the high compactness of the film controls the size-exclusion mechanism towards interferent molecules, irrespective of their charged state, thus broadening the spectrum of compounds that can potentially be rejected by the polymer. Remarkable selectivity was therefore achieved, along with satisfactory sensitivity and precision, encouraging the employment of the device for DA detection in human urine samples with good recovery.

## 2. Materials and Methods

### 2.1. Reagents and Sample

*O*-aminophenol (99%), dopamine (99%), ascorbic acid (99%), uric acid (99%), tyramine (98%), and serotonin (98%) were purchased from Sigma-Aldrich (Germany). All other chemicals were of analytical reagent grade. Prior to its use, *o*-aminophenol was recrystallized in ethyl acetate for its purification [26].

All the solutions were prepared by employing pure water supplied by a Milli-Q RG unit (Millipore, Bedford, MA, USA). Solutions from the species susceptible to oxidation, such as *o*-aminophenol, dopamine, and interferents, were prepared just before their use.

A healthy adult volunteer provided the urine sample, which was centrifuged for 10 min at 8000 rpm, and then diluted 100-fold with phosphate buffer (PBS) at pH 7 before analysis. Donor gave his consent to the use of the urine sample after being adequately informed. The employment of a biological sample for research purposes was authorized by the Department of Sciences according with the Ethics Committee of the University of Basilicata.

### 2.2. Apparatus

Cyclic voltammetry experiments were performed using a 263A potentiostat/galvanostat from EG&G (Princeton Applied Research, Princeton, NJ). The M270 software version 4.23 (EG&G) was employed for data acquisition. Differential pulse voltammetry (DPV) experiments were carried out with a CHI660B potentiostat (Shanghai CH Instrument Company, Shanghai, China).

A standard three-electrode cell was used throughout the work, consisting of a saturated calomel reference electrode (SCE), a platinum wire counter electrode, and a platinum working electrode made of a 3 mm disk inserted into a PTFE cylindric frame.

A Seiko QCA 917 quartz crystal microbalance (EG&G) was used to perform the gravimetric experiments, equipped with the WinEchem software version 2.0 for data acquisition. A glass bottom cell model RG100 was mounted on the top of a QA-CL4 Well-type Teflon quartz crystal holder (EG&G). A 9 MHz AT-cut quartz crystal (EG&G) was used, covered on both sides by platinum disks of a 5 mm diameter.

Electrochemical and gravimetric experiments were carried out at room temperature.

### 2.3. Fabrication of the MIP Sensors

A Pt working electrode was cleaned following an optimized procedure consisting of three steps [27]. Firstly, the electrode was mechanically polished by abrasion with alumina (0.05 μm particles) and sonicated in double distilled water. Then, it was sonicated for a few minutes in hot nitric acid (70%) for chemical cleaning. Finally, it was electrochemically treated by cycling in 0.5 M sulfuric acid in the potential range of $-0.225V/+1.25$ V (vs. SCE) at a scan rate of 100 mV/s for the number of cycles sufficient to get a steady state current profile. After rinsing with double distilled water, the electrode was ready to be modified.

The MIPs were realized on the Pt electrode through PoAP electrosynthesis by cyclic voltammetry in the presence of dopamine. The potential was varied in the interval from $-0.1$ to +0.9 V (vs. SCE) at a scan rate of 100 mV/s for 30 scan cycles. A deposition solution of 5 mM oAP and 20 mM DA in an acetate buffer of 0.1 M at pH 5.0 was employed. The modified electrode was then washed with pure water before its use.

The extraction of the DA template molecules entrapped within the polymer was carried out by immersing the modified electrode in a solution of 0.5 M sulfuric acid for 1 h and 15 min. Then, the electrode was rinsed with double distilled water and incubated for 20 min in DA solutions of increasing concentrations prepared in a 0.1 M PBS at pH 7. This allowed analyte recognition and rebinding into the MIPs cavities left empty after the extraction.

### 2.4. Electroanalytical Measurements

DA extraction from the MIP was verified by employing a solution of 5 mM potassium ferricyanide in $KNO_3$ 0.1 M as an electrochemical probe. The voltametric profile of potassium ferricyanide was acquired before and after the extraction in the potential range from 0.6 to $-0.2V$ at a scan rate of 50 mV/s. In the same potential interval, DPV experiments were also performed on the ferricyanide solution, employing a pulse amplitude, pulse period, and pulse width of 50 mV, 0.2 s, and 50 ms, respectively. The potential increment was 4 mV, and the sampling width was 20 ms.

DA detection was carried out by DPV in a PBS at pH 7 as a supporting electrolyte. Measurements were done in the potential range from $-0.1$ to +0.4 V, unless otherwise stated, employing the same parameters previously reported for the extraction.

All the CV and DPV experiments were shown in the relevant figures by setting the anodic currents downward (negative scale) and the cathodic currents upward (positive scale).

### 3. Results

#### 3.1. Electrochemical Behavior of DA at Pt Electrodes: Influence of pH

Our previous studies carried out on PoAP evidenced that the employment of a neutral pH during electrosynthesis assures the realization of an insulating polymer [25], whereas

in acid media, a conducting film is obtained following a completely different growth mechanism [28]. As it was previously highlighted, the availability of the polymer in the insulating form is an important requirement to ensure adequate selectivity to the resulting MIP.

Moreover, the feasibility of using a neutral pH for MIP electrodeposition was explored. With this aim, a preliminary investigation on the electrochemical behavior of DA at Pt electrodes was carried out by cyclic voltammetry. The voltammetric profile of DA at a concentration of 2.5 mM in a PBS at pH 7 acquired by varying the scan rate is shown in Figure 1. At lower scan rates, only the anodic peak is present, whereas the cathodic counterpart begins to be evident from 50 mV/s. This finding indicates that DA oxidation follows an electrochemical/chemical (EC) mechanism: DA oxidizes to dopamine-o-quinone which, upon deprotonation, undergoes a chemical cyclization process giving leucodopaminechrome through a 1,4 Michael addition (see Scheme S1 in the Supporting Information) [23]. At a lower scan rate, the time scale of the electrochemical process is sufficiently slow to allow the chemical process to occur to a higher extent, and then only the oxidation peak is evident. Indeed, when diagnostic tests for the EC mechanism were performed [29], as illustrated in Figure S1 of the Supporting Materials, they were only partially fulfilled by employing the voltammetric parameters evaluated from Figure 1. This would suggest the onset of complications in the electron transfer, such as electrode poisoning by DA or its oxidation products.

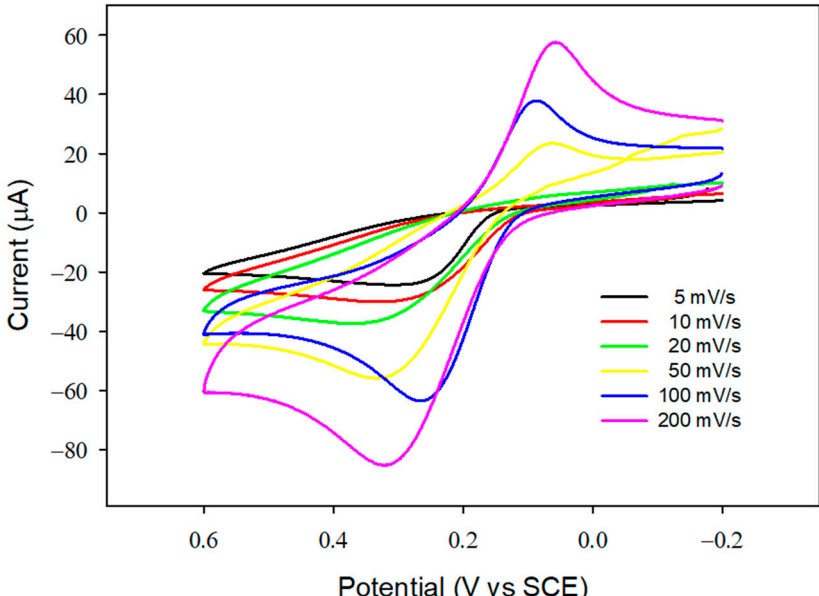

**Figure 1.** Cyclic voltammograms relevant to 2.5 mM of dopamine in phosphate buffer solution (I = 0.1 M, pH 7) at a bare platinum electrode acquired by varying the scan rate in the range of 5–200 mV/s.

The voltammetric profile of DA at a concentration of 2.5 mM in a PBS at pH 7 acquired upon continuous cycling with a scan rate of 10 mV/s is reported in Figure 2a. As can be seen, a progressive current decrease occurs with the scan cycles until reaching a flat profile, indicative of the passivation of the electrode surface by the DA oxidation products. This behavior is typically found during the electrosynthesis of insulating films, such as polyphenols, and is caused by the formation of oligomers and, therefore, of insoluble polymers that precipitate on the electrode surface. Such a complication was also observed at faster scan rates.

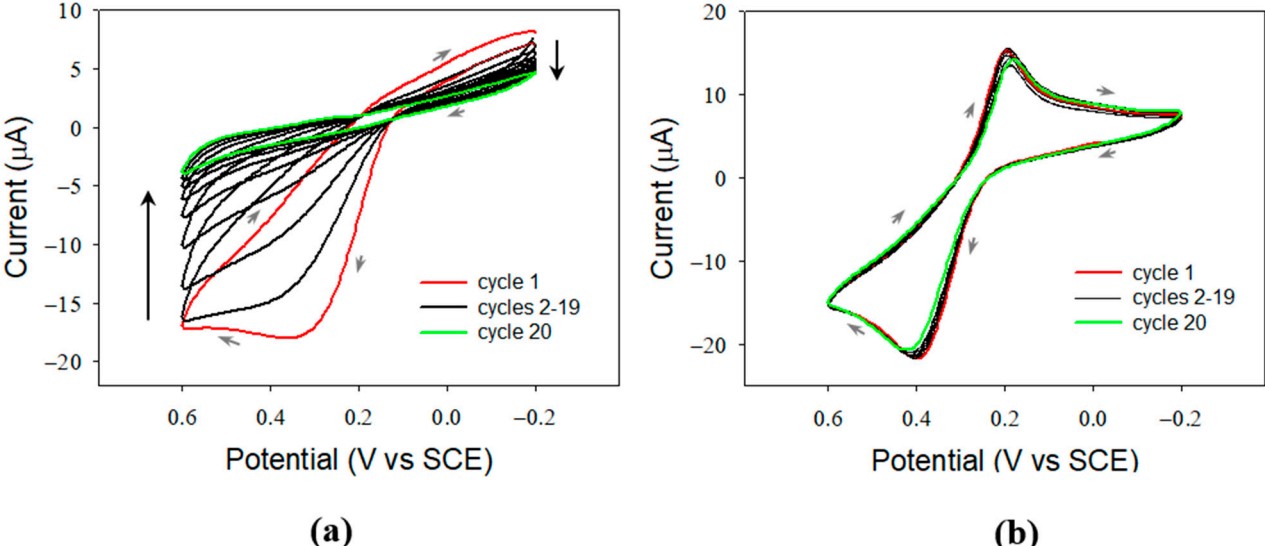

**Figure 2.** Cyclic voltammograms (20 cycles) relevant to dopamine 2.5 mM in phosphate buffer solution (I = 0.1 M, pH 7) (**a**) and in acetate buffer solution (I = 0.1 M, pH 5) (**b**) at a bare platinum electrode. Scan rate: 10 mV/s. The black arrows indicate the current evolution upon subsequent cycles, whereas the grey arrows indicate the scan direction.

To quantify the extent of the electrode fouling, differential pulse voltammetry was used. Two DPV experiments were performed consecutively using dopamine solutions with the same concentration and pH. The potential was scanned in the range of −0.1/0.6V, obtaining the current profiles shown in Figure S2A. The effect of the poisoning is evidenced by both the shift of the peak potential of about 60 mV towards more anodic values, and the decrease in the peak current of about 48%. In order to verify if the extent of the electrode poisoning could be related to the DA concentration in the solution, the experiment was repeated using a concentration of 0.1 mM (Figure S2B). The decrease in concentration significantly attenuates electrode poisoning since the positive shift of the peak potential was reduced to 4 mV and the current attenuation was only 4.9%. The suggestion is, therefore, to employ DA concentrations no higher than 0.1 mM, which are clearly lower than the value of 5 mM employed for *o*-aminophenol to synthesize the polymer. A reduced trapping of the template molecules in the MIP would then result.

An alternative mean to prevent dopamine polymerization comes from the investigation of its oxidation route [23]. As is shown in Scheme S1, leucodopaminechrome, formed upon cyclization, can react with dopamine-o-quinone to generate 5,6-dihydroxy-indolin-quinone (dopamine chrome) which, in turn, is responsible for a cascade of processes that ultimately lead to the formation of a polymeric structure that precipitates on the electrode surface. More precisely, dopaminechrome is first isomerized to 5,6-dihydroxylindole, then oxidized to 5,6-indoloquinone which, in turn, leads to poly (5,6-indoloquinone) [30]. The key step for the polymer formation is the deprotonation of the oxidized dopamine, which is necessary for cyclization to take place. Since deprotonation occurs at a pH > 5, working at pH 5 cyclization is prevented, as well as electrode passivation, even at a DA concentration as high as 2.5 mM. As it is possible to see in Figure 2B, where the same voltammetric experiment carried out at pH 7 is reported, at pH 5, both peaks are evident despite the low scan rate, and peak intensities remain stable by continuous cycling.

In light of these considerations, a solution of 5 mM OAP and 2.5 mM DA in acetate buffer, at a pH of 5 and an ionic strength 0.1 M, was used for the electrosynthesis of the MIP. At a mildly acidic pH, PoAP still grows as insulating film.

### 3.2. Evaluation of the Voltammetric Parameters for MIP Electrosynthesis

Once the pH value of the supporting electrolyte was defined, an optimization study of the voltammetric parameters used for MIP electrodeposition was conducted. The starting

conditions were those already optimized for PoAP electrosynthesis at pH 7 [25]: 20 scan cycles at 50 mV/s in the potential range of $-0.1$ V/$+0.9$ V vs SCE. The first parameter investigated was the number of scanning cycles, which was gradually increased from 20 to 30 and then to 40.

The effectiveness of entrapping DA by varying the electrosynthesis conditions was evaluated indirectly, using a potassium ferricyanide electrochemical probe. Its voltametric profile on platinum abates after the MIP deposition, and is partially restored upon DA extraction by immersing the modified electrode in a solution of 0.5 M $H_2SO_4$ for 1 h. The extent of the current increase upon extraction with respect to the basal value registered soon after MIP deposition, indicated as $\Delta i$, was computed. As is possible to see in Figure S3, this parameter was raised, going from 20 to 30 scan cycles, and then decreased, going to 40 scan cycles. Increasing the number of cycles involves, firstly, a greater amount of the template trapped within the polymer matrix. However, if it is further increased, the MIP obtained has a structure so compact as to prevent the release of the template molecules in the solution.

Then, the scan rate was varied. Going from 50 mV/s to 100 mV/s, at a fixed scan number of 30, $\Delta i$ slightly increased. It is important to stress the convenience of working at higher scan rates also in relation to the reduced tendency of DA oxidation products to cyclize.

From this point on, the MIP deposition was carried out by scanning the potential in the range of $-0.1/0.9$ V vs SCE for 30 cycles at a scan rate of 100 mV/s.

### 3.3. Evaluation of the Monomer to Template Concentration Ratio

In the realization of an MIP, the number of cavities that remain shaped in the polymeric matrix is strongly influenced by the concentration ratio between the monomer and the template. As concerns the monomer, oAP, a concentration of 5 mM was employed in accordance with the value reported in our previous studies on PoAP carried out at pH 7 [25]. A further increase of monomer concentration was not feasible because of its poor solubility in aqueous solutions close to neutrality. Indeed, suspensions prepared at higher concentrations were left under sonication for prolonged times to favor their complete dissolution. In these conditions, however, monomer oxidation occurs at an appreciable extent, testified to by the intense yellow color assumed by the solution. In order to optimize the monomer to template ratio, only the DA concentration was varied while keeping that of the oAP fixed at 5 mM.

Figure 3 shows the current variation $\Delta i$, calculated as previously explained, related to the MIPs deposited by varying the concentration of the trapped analyte.

As the concentration of DA in the deposition solution increases, the number of template molecules retained inside the MIP increases and, consequently, so does the number of cavities left empty upon the acidic extraction. The trend of the current variation becomes almost stationary for concentration values close to 20 mM, which were then selected as the optimal DA concentrations. This means that a molar concentration ratio of 1:4 between the monomer and the template (oAP: DA) was proposed for the MIP deposition.

### 3.4. MIP Deposition on Pt and Its Molecular Recognition

The voltammetric profile, acquired during MIP electrosynthesis following the experimental conditions previously optimized, is reported in Figure 4A. The attenuation of the oxidation current until reaching a flat profile is typical of the formation of an insulating polymer, as already discussed [25]. A fuller understanding of the current signal relevant to the first cycle was possible by comparing the voltammetric profiles registered at the same experimental conditions in the presence of only the monomer oAP (Figure 4B), and only the template DA (Figure 4C).

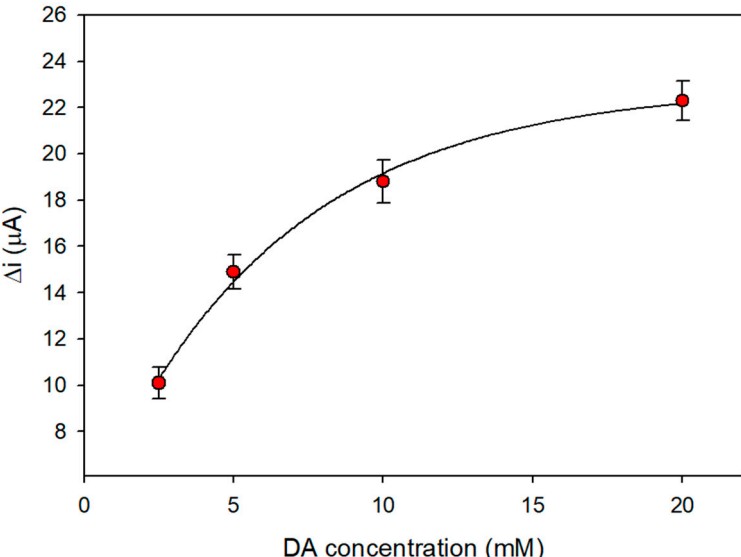

**Figure 3.** Influence of DA concentration in the deposition solution on MIP performances. Each value represents the mean of triplicate measurements; the error bars represent the relevant standard deviations. MIP deposition was carried out by scanning the potential in the range −0.1/0.9 V vs SCE for 30 cycles at a scan rate of 100 mV/s, employing a 5 mM monomer solution in acetate buffer at pH 5.0.

oAP oxidation generates a current peak at about 0.33 V (Figure 4B), which is also present in the MIP profile, being slightly shifted to about 0.36 V and having almost the same intensity. These findings indicate that the presence of DA in the deposition solution, even at a higher concentration, does not influence monomer oxidation and polymer formation.

In the MIP profile, there is an additional current contribution at about 0.53 V, which can be assigned to DA. As a confirmation of such an attribution, in the voltammetric profile shown in Figure 4C, DA oxidation occurs at the same potential value. With subsequent scans, a shift towards a more positive value of 0.66 V is noticed, probably due to DA adsorption on the electrode surface. The employment of pH 5 prevents DA polymerization, as just a moderate current decrease is noticed without reaching a flat profile. The extremely high concentration, however, causes moderate electrode fouling, testified to by the shift in the potential peak.

It is worth noting that this phenomenon is hindered when oAP is also present in the solution. Indeed, in the MIP profile (Figure 4A), the DA contribution lies at 0.53 V, and then at the starting oxidation potential of Figure 4C, before poisoning occurs. Moreover, the intensity of such a contribution is markedly smaller than that registered in the absence of oAP (note the different current scale of Figure 4A,C). This means that during the first forward half-cycle, PoAP formation has already occurred to an extent sufficient to allow DA oxidation on the Pt zones still uncovered, while preventing fouling problems.

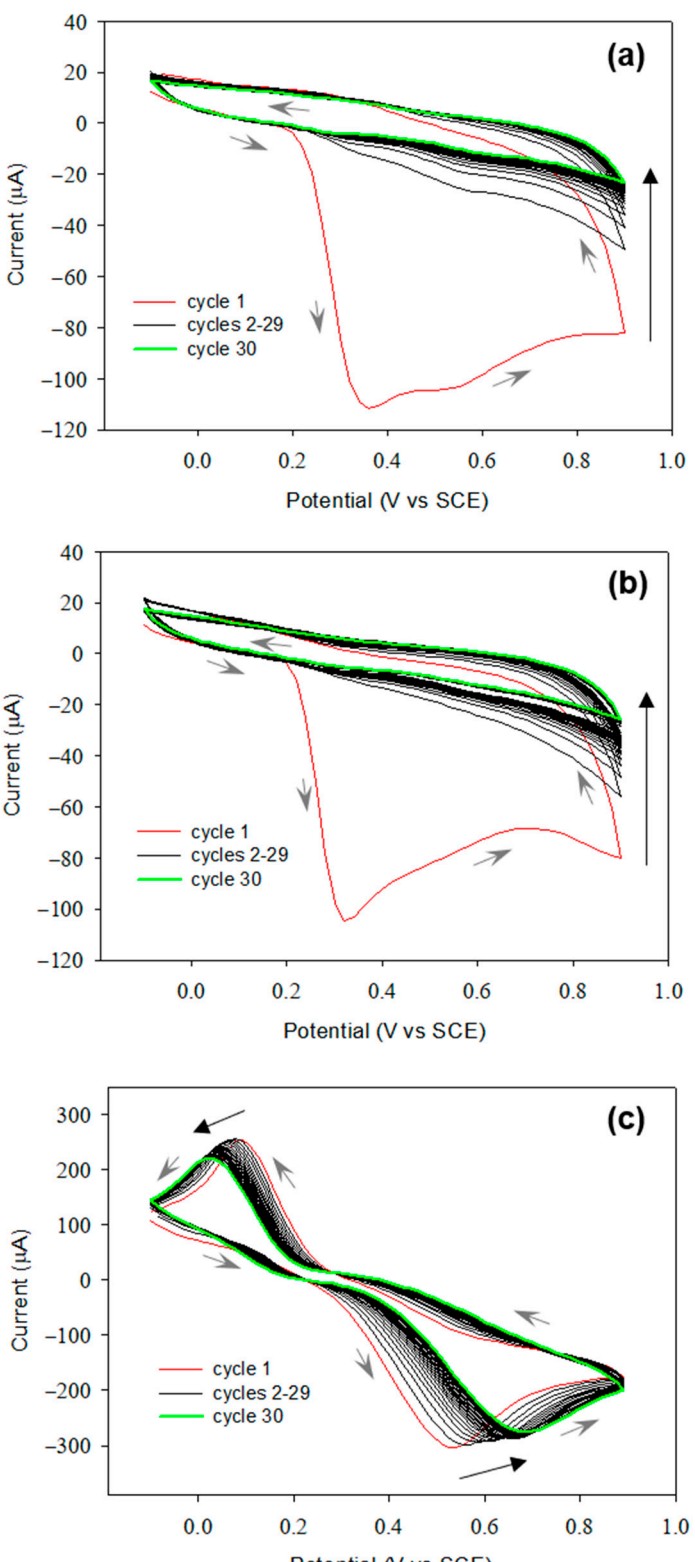

**Figure 4.** Cyclic voltammograms relevant to 5mM oAP and 20mM dopamine (**a**); 5 mM oAP (**b**); 20 mM dopamine (**c**) in acetate buffer solution (I = 0.1 M, pH 5), at a bare platinum electrode acquired at 100 mV/s by scanning the potential in the range −0.1/0.9 V for 30 cycles. The black arrows indicate the current evolution upon subsequent cycles, whereas the grey arrows indicate the scan direction.

The ability of the MIP to entrap the target analyte used as a template during the deposition was evaluated. The experiment was carried out employing a 5 mM potassium

ferricyanide solution by cyclic voltammetry (Figure 5A), and by DPV (Figure 5B). Ferricyanide gives an intense current signal at bare Pt which, upon MIP electrosynthesis, is flattened. The electron transfer was blocked by the nonconducting poly-o-aminophenol membrane covering the electrode. After leaving the modified electrode immersed in a 0.5 M sulfuric acid solution for 1h, the current signal was partially restored since ferricyanide molecules can permeate the cavities of the MIP left empty after the extraction. Then, if the MIP is immersed in a solution of 0.1 mM DA for 30 min, an attenuation of the ferricyanide current signal is observed, justifiable in terms of the entrapment of the DA in the cavities previously formed. For the analyte entrapment, a DA solution at pH 7 was used. The neutral pH did not cause any passivation of the electrode surface from the analyte. Indeed, incubation took place without applying a potential to the electrode, thereby avoiding the analyte oxidization that activates polymerization. Furthermore, as we have previously evidenced, at concentration values no higher than 0.1 mM, even if an oxidation potential is applied, polymerization does not occur when working at pH 7.

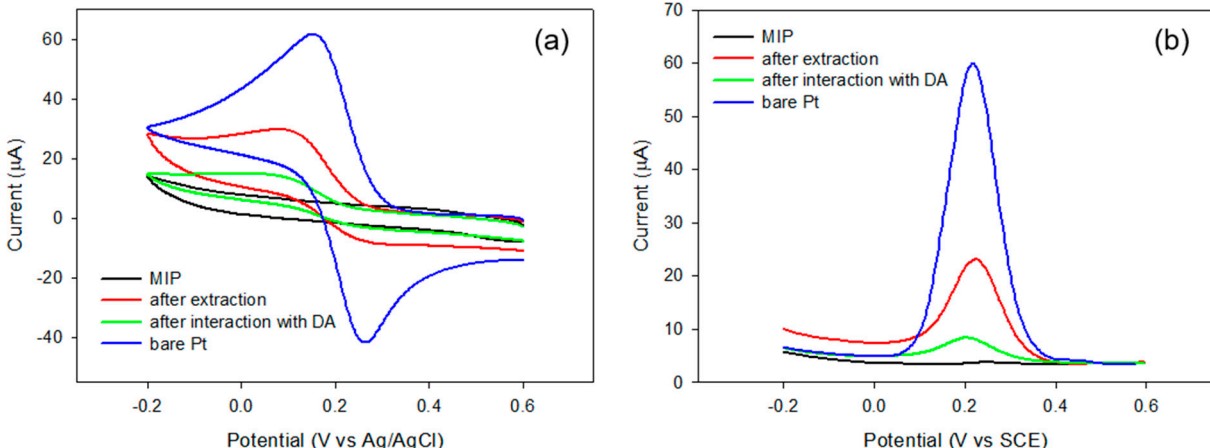

**Figure 5.** (**a**) Cyclic voltammograms of 5 mM of ferricyanide on a bare Pt electrode (blue profile), on an MIP-modified Pt electrode, before (black profile), and after (red profile) removal of the imprinted DA molecules, and on an MIP-modified Pt electrode after interaction with 0.1 mM of DA (green profile). Scan rate 50 mV/s. (**b**) DPV profile of 5mM of ferricyanide acquired in the same conditions of the cyclic voltammograms. Potential range from +0.6 V to −0.2 V. Potential increment: 4 mV; sampling width: 20 ms; pulse amplitude: 50 mV; pulse period: 0.2 s; pulse width: 50 ms.

The most suitable solvent to efficiently extract DA molecules from the MIP, without altering the compactness of the PoAP polymeric matrix, was evaluated. As an alternative to sulfuric acid, the employment of ethanol as an extraction solvent was tested. The entrapment of DA within PoAP occurs through H-bond interactions between the phenolic hydroxyl groups, or the amine groups, in DA, and the nitrogen and oxygen heteroatoms of the PoAP network, as it is shown in Scheme 1. We have previously demonstrated the presence of water molecules H-bonded within the polymer that sustain polymer chain alignment vertically on the platinum substrate [25]. Template molecules probably compete with water or give rise to additional H-bonds. Acid and alcohol are useful solvents for template extraction: DA molecules are released from the imprinting membranes because of the destruction of H-bonds in strongly acidic conditions; on the other hand, additional H-bonds can form between the alcohol and the template, which might compete with those between the monomer and the template, resulting in a reduction in binding energy between the template and the monomer.

**Scheme 1.** Schematic representation of the possible interactions between the template molecules and the imprinted cavities within the MIP. PoAP functionalities are reported according to the structure derived in our previous studies [25].

The extraction with ethanol was carried out by leaving the MIP immersed in pure alcohol for 20 min, and its effectiveness was evaluated by monitoring the variation in the voltammetric profile of potassium ferricyanide. After extraction, an increase of the current signal was observed that, however, did not attenuate upon sensor incubation with DA. A PoAP degradation in organic media can justify the high current recorded after the extraction. Indeed, if the same experiment is carried out using the NIP (non-imprinted polymer), PoAP permanence in ethanol causes a comparable increase in the ferricyanide current. This experimental evidence indicates that the immersion in pure ethanol did not help to release dopamine from the complementary cavities of the MIP, but simply degraded the polymer. Conversely, if the NIP is immersed in sulfuric acid for one hour, the current signal recorded is negligible, highlighting an appreciable PoAP stability in acidic media. Therefore, 0.5 M of sulfuric acid was employed as an extraction solvent.

Finally, the extraction and trapping times of the template inside the MIP was optimized by employing a quartz crystal microbalance (QCM), which allowed us to directly evaluate these two parameters through one-step experiments. Precisely, the variation of the resonant frequency of a quartz crystal covered by a platinum electrode modified with the MIP was monitored during the extraction and the entrapment steps. The gravimetric profiles are illustrated in Figure S4.

During extraction, the frequency profile initially increases, reaching a plateau starting from about 4000 s. The optimal time for the extraction was set at 4500 s (1h and 15 min) to ensure that the stationary region had been reached. A similar study was conducted during the entrapment by immersing the MIP electrode in a solution of 0.1 mM of DA at pH 7. The acquired gravimetric profile, illustrated in Figure S4B, initially decreases, as expected, then becomes stationary when the dopamine molecules present in the solution have filled the available cavities present in the polymer. An entrapment time of 1200 s (20 min) was selected.

*3.5. Analytical Characterization of the MIP*

The analytical performances of the MIP sensor were evaluated by carrying out a calibration curve at different DA concentrations. The direct electrochemical detection of DA by DPV was used. The current peaks acquired are shown in Figure 6a. The current increases as the DA concentration in the incubation solution is increased. Much lower current signals were registered with respect to the DPV profiles, reported in Figure 5b. At this point, it is worth noting that a different operating mode was adopted to carry out the experiments reported in Figures 5b and 6a: ferricyanide current profiles were generated from an electrochemical probe present in the solution, whereas the dopamine current profiles were generated from an electrochemical probe trapped within the polymeric

membrane immobilized onto the electrode surface. The different concentration levels employed (millimolar versus micromolar) should also be noted.

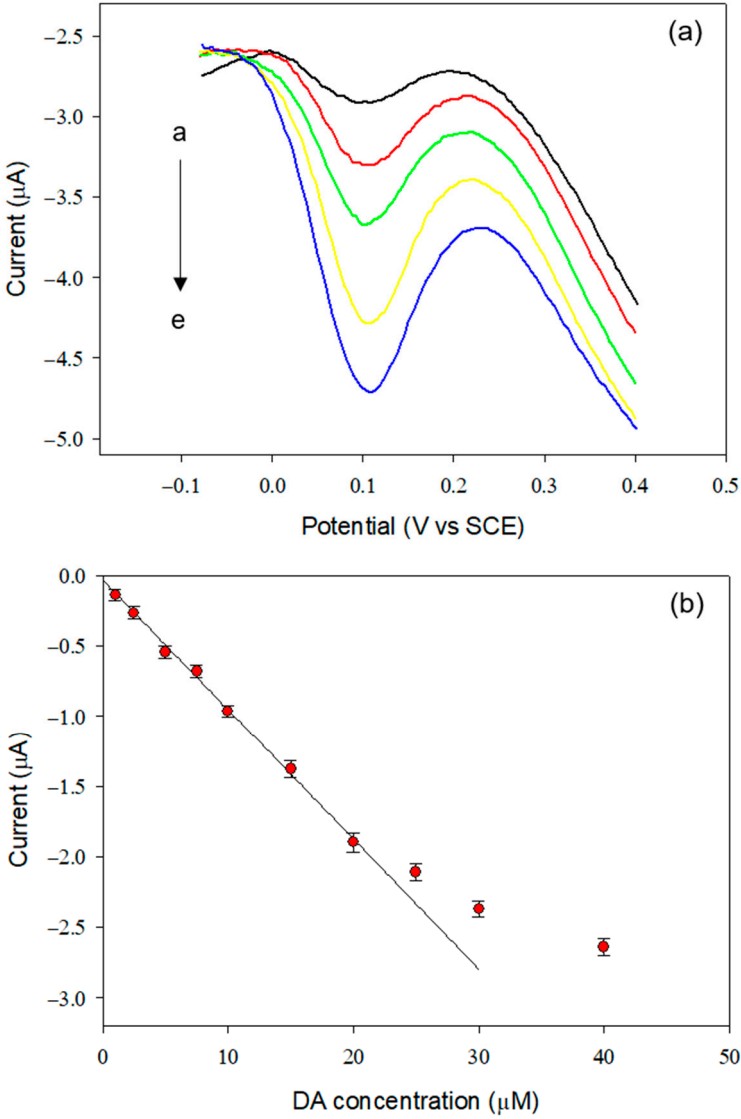

**Figure 6.** (**a**) DPVs of dopamine at a concentration range of 2.5–15 μM (curves a–e) at the MIP-modified electrode. Electrolyte: phosphate buffer solution, pH 7.0 and I = 0.1 M. Potential range from −0.1 V to +0.4 V. Potential increment: 4 mV; sampling width: 20 ms; pulse amplitude: 50 mV; pulse period: 0.2 s; pulse width: 50 ms. (**b**) Calibration curve. Each value represents the mean of triplicate measurements; the error bars represent the relevant standard deviations.

The relevant calibration curve is reported in Figure 6b. A linear range up to 20 μM was achieved. From the linear regression equation ($r^2$ = 0.9977), a sensitivity of $(-9.2 \pm 0.2) \times 10^{-2}$ μA/μM, and a limit of detection, LOD, of 0.65 μM [31], were evaluated.

The LOD value achieved was satisfactory considering that a conventional platinum electrode was used. Indeed, several MIPs for DA realized on composite electrodes made of nanomaterials are reported in which, despite the enhanced electrode area, limits of detection of the same order have been obtained. To cite some examples, in the case of a modified GC electrode with nanoporous gold leaves [32], the detection limit was 0.3 μM. Wang and coworkers [33] have recently realized a novel dopamine imprinted chitosan/CuCo2O4@carbon/three-dimensional macroporous carbon-integrated electrode which allows for the obtainment of an LOD of 0.16 μM requiring, however, an extremely complex procedure for sensor preparation. Conversely, in this work, we propose an

efficient one-step procedure for realizing the MIP within just ten minutes. For a glassy carbon electrode (GCE) modified with graphene films, molecularly imprinted over-oxidized poly(pyrrole), and electrochemically deposited gold nanoparticles, the detection limit was estimated to be 0.1 µM and, again, a more complicated preparation was adopted [34]. Elsewhere, the employment of composite nanomaterials allowed for the achievement of LODs significantly lower, with nanomolar [35], or even picomolar [36], concentration values.

Among the MIPs realized on bare electrodes, an LOD of 1.04 µM was obtained on gold modified with poly(pyrrole) [37]. Lower detection limits in the nanomolar range were achieved in the case of poly(ethacridine) [38], and poly(pyrrole–phenylboronic acid) [39], both deposited on glassy carbon. In the first case, the sensor was applied for DA determination in human sera with good recoveries, even if no data were displayed on its selectivity. In the second case, the sensor proved to be selective for DA and was employed for the analysis of dopamine hydrochloride injections. Its effectiveness for the analysis of more complex samples, such as biological fluids, was not demonstrated.

As was anticipated, there is only one report concerning an MIP for DA on Pt electrodes [20]. In this case, a limit of detection as low as 4.5 nM was detected. It is worth noting that the differential pulse voltammograms shown in the paper evidenced current signals for DA lower that that reported in the present work at the same concentration values. The limit of detection was evaluated by chronoamperometry, at a fixed potential of 0.12 V, adding aliquots of a standard DA solution in the cell. The different experimental conditions adopted do not allow a direct comparison between the two sensors. Moreover, the DA's rebinding performance relative to ascorbic acid was only 2.7-fold, evidencing a moderate MIP selectivity towards dopamine. Finally, as was already discussed in the Section 1, the procedure employed for MIP realization was more complex, being based on a first step for poly(pyrrole) electrosynthesis, and a second step for the overoxidation necessary to achieve adequate permselectivity. The experimental conditions adopted to carry out the two steps were not as mild as those employed for PoAP electrosynthesis and, to minimize DA oxidation and polymerization, degassing solutions were necessary.

The reproducibility of the sensor was investigated using a DA concentration of 10 µM. For five electrodes prepared in the same way, an acceptable RSD of 6.2% was obtained indicating a good electrode-to electrode reproducibility of the fabrication method. Using the same electrode, the within-day and day-to-day relative standard deviations for replicate measurements (n = 5) carried out employing a 10 µM DA solution were 4.7% and 5.8%, respectively.

After each use, the electrode was stored at room temperature. The current response decreased by about 20% after one month, compared to its initial response, evidencing a relatively acceptable stability. The satisfying stability is derived from the mechanical/chemical stability of MIP compared with natural receptors (e.g., antibodies, enzymes).

### 3.6. Selectivity of the MIP

Molecular imprinting technology is based on the high affinity of the MIP cavities towards the template molecules. The recognition mechanism of the analyte of interest, based on the "lock and key" model, should prevent the entrapment of any other substance present in the real sample analyzed. Selectivity was evaluated towards competitive molecules, including the structural analogues of DA, tyramine and serotonin, and interferents coexisting in biosamples, such as ascorbic acid and uric acid (their structures are given in Table 1). A preliminary investigation was conducted to verify the electroactivity of these substances on a conventional platinum electrode. DPV signals in the potential range of −0.2 V/+ 0.6 V were acquired using a phosphate buffer at pH 7, and an ionic strength of 0.1 M as a supporting electrolyte. The peak potential values of the interferents and of DA are reported in Table 1.

**Table 1.** Oxidation potentials of dopamine and possible competitive molecules at a bare Pt electrode.

| Compound | Structure | Oxidation Potential [1] |
|---|---|---|
| Dopamine (DA) | | +0.135 V |
| Ascorbic Acid (AA) | | +0.289 V |
| Uric Acid (UA) | | +0.328 V |
| Serotonin (SER) | | — |
| Tyramine (TYR) | | +0.310 V |

[1] Oxidation potentials were evaluated by DPV in phosphate buffer solutions (pH 7.0, I = 0.1 M).

Dopamine peak potential at pH 7 is at +0.135 V. In the same pH conditions, tyramine, ascorbic acid, and uric acid are also electroactive, and their peak potentials are very close to each other but far enough from that of dopamine. Serotonin, on the other hand, did not show a net current peak. Therefore, no faradic interference is to be expected by setting the detection potential at the oxidation value of DA. In order to evaluate the sensor selectivity, the same experiment was carried out on the NIP (non-imprinted polymer), i.e., on PoAP electrosynthesized in the absence of DA in the deposition solution, and on the MIP. DPV peaks were acquired after incubating the sensor with solutions of 10 µM of DA and interferents at a given concentration. Particularly, a concentration of 50 µM was used for serotonin and tyramine, whereas 100 µM was the concentration used for ascorbic acid and uric acid. The results are shown in Figure S5. It should be noted that the current values have been measured at the different potentials, reported in Table 1. Serotonin gave no response on both the electrodes, as already experimented on bare Pt. The NIP current signals from all the other tested compounds are negligible because of the compactness of PoAP. When the MIP was incubated in the interferent solutions, a minimal entrapment was noticed, causing a current signal that, in the worst case (AA), was only about 20% of that of DA.

Then, to confirm the MIP selectivity towards DA, mixtures of DA and each of the interferents at the concentration values of Figure S4 were analyzed. The DPV peaks acquired after incubating the sensor with the various solutions are illustrated in Figure 7a.

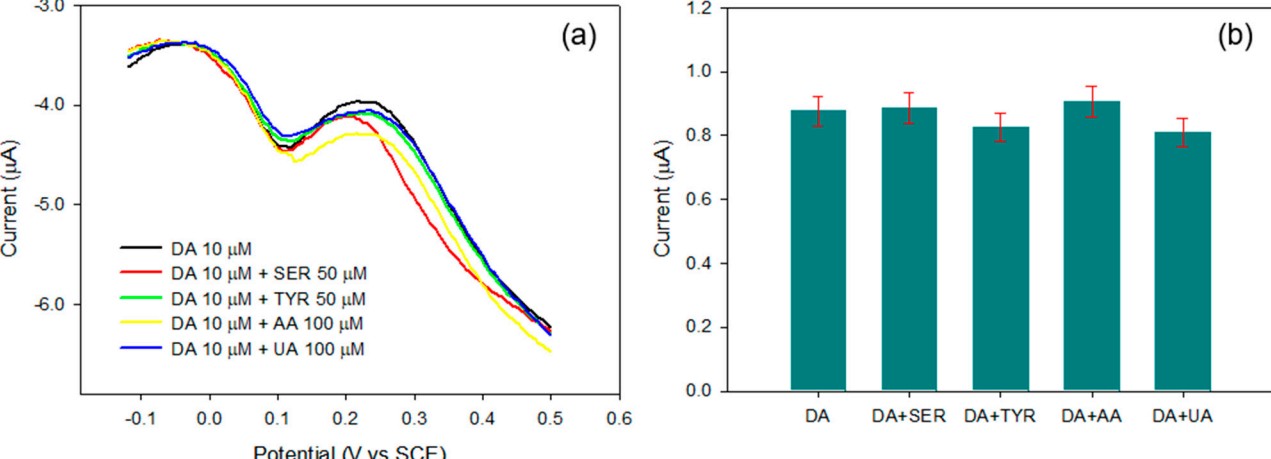

**Figure 7.** (**a**) DPVs of 10 μM dopamine in the absence and in the presence of serotonin 50 μM, tyramine 50 μM, ascorbic acid 100 μM, and uric acid 100 μM at the MIP-modified electrode. Electrolyte: phosphate buffer solution, pH 7.0 and I = 0.1 M. Potential increment: 4 mV; sampling width: 20 ms; pulse amplitude: 50 mV; pulse period: 0.2 s; pulse width: 50 ms. (**b**) Mean values and error bars relevant to replicate experiments (n = 3).

As can be seen from the figure, only the dopamine oxidation peak is evident in the potential range considered. Despite the presence of the interferents in the incubation solution, additional peaks were not generated, and the intensity of the DA peak does not significantly change. Indeed, as it is reported in Figure 7b, if the experiment is repeated in triplicate, the current signals obtained in the presence of the interferents are not significantly different from those obtained employing only dopamine within the limits of the method reproducibility. The MIP electrosynthesized on Pt is, therefore, highly selective: the cavities formed upon extraction show an affinity only towards dopamine, as is attested by the reproducibility of the of DA peaks. Moreover, apart from the imprinting cavities, the general film compactness hinders any permeation of the interferents to the electrode surface, and then no additional peaks are present.

### 3.7. Real Sample Analysis

In order to examine the reliability of the proposed sensor in practical applications, it was employed for DA detection in a human urine sample. The experiments were performed in triplicate and recovery tests were conducted via the standard addition method. The results are illustrated in Table 2. The obtained recoveries ranged from 92.5% to 109.1%, thereby validating the accuracy and practicability of the developed sensor.

**Table 2.** Determination of DA in human urine samples.

| Added (μM) | Found (μM) | Recovery (%) | RSD (%) [1] |
| --- | --- | --- | --- |
| 0 | n.d [2] | - | - |
| 4 | 3.70 | 92.5 | 5.1 |
| 8 | 8.73 | 109.1 | 3.9 |

[1] RSD values reported are for n = 3. [2] Not detected.

### 4. Conclusions

A novel amperometric sensor has been proposed for dopamine detection based on a molecularly imprinted PoAP film electrosynthesized on a Pt electrode. The employment of Pt for the realization of MIPs for DA is unusual because of its tendency towards fouling upon the absorption of the oxidative products, even polymers. Indeed, DA oxidation is complicated by a chemical cyclization step with the subsequent formation of polymeric structures precipitating from the solution to the electrode surface. A preliminary investigation on the electrochemical behavior of Pt, along with a careful evaluation of

the experimental conditions to be used for the MIP realization, allowed us to successfully overcome these limitations. The pH solution proved to be a key factor in preventing DA polymerization, even at high concentrations. PoAP deposition occurred while avoiding any DA involvement. Hence, the template molecules could be easily extracted from the MIP by immersion in an acidic solution, and then trapped within the imprinted cavities with an appreciable reproducibility and notable selectivity. No interference was registered from the electroactive compounds usually present in biological samples, or from species with a structure similar to DA and that are, therefore, potential competitors during the entrapment stage within the MIP cavities. The reliability of the device in practical applications was demonstrated by analyzing human urine samples. Good recovery percentages were achieved. Despite the fact that the limit of detection was not particularly low, the overall performances of the MIP, and the fast and easy procedure designed for its preparation, encourage its future employment for DA quantification in biological samples from patients suffering from pathologies involving an increase in the physiological values of this important analyte. Work in this direction is in progress in our laboratory.

**Supplementary Materials:** The following are available online at https://www.mdpi.com/article/10.3390/chemosensors9100280/s1, Scheme S1: Schematic representation of the electrochemical and chemical processes involved upon dopamine oxidation, causing the formation of poly(dopamine). Figure S1: Diagnostic tests for Electrochemical-Chemical (EC) reactions. (a) Test 1: $|Ip_C/Ip_A|$ is less than one but tends to unity as the scan rate is increased, where $Ip_C$ and $Ip_A$ are the cathodic and anodic peak currents, respectively. The test has been verified from 50 mV/s since, at lower scan rates, the cathodic peak is not present, and is satisfied. (b) Test 2: $Ip_A/v^{1/2}$ decreases slightly with the increasing scan rate. The test is satisfied in the whole range of scan rates investigated. (c) Anodic peak potential shifts positively with increasing scan rate. The test is satisfied only at lower scan rates. Figure S2: DPV profiles of a dopamine solution 2.5 mM (a) and 0.1 mM (b) in phosphate buffer pH 7.0, I 0.1 M, acquired on a platinum electrode during two successive experiments. Potential increment: 4 mV; sampling width: 20 ms; pulse amplitude: 50 mV; pulse period: 0.2 s; pulse width: 50 ms. Figure S3: Influence of the scan cycles on MIP performances. Each value represents the mean of triplicate measurements; the error bars represent the relevant standard deviations. MIP deposition was carried out by scanning the potential in the range $-0.1/0.9$ V vs SCE at a scan rate of 50 mV/s, employing a 5 mM monomer and 2.5 mM dopamine solution in acetate buffer at pH 5.0. Figure S4: (a) Frequency variation recorded by QCM during dopamine extraction from the MIP by immersion in a solution of $H_2SO_4$ 0.5 M. (b) Frequency variation recorded by QCM during dopamine entrapment by immersing the MIP in a 0.1 mM dopamine solution in phosphate buffer at pH 7.0. Figure S5. DPV current signals relevant to 10 μM DA, 50 μM TYR, 100 μM AA, and 100 μM UA at the NIP and at the MIP. The NIP, not imprinted polymer, was electrosynthesized employing the same experimental conditions as Figure 4a in the absence of DA. Each compound current signal was measured at the oxidation potential indicated in Table 1.

**Author Contributions:** Conceptualization, R.C.; Funding Acquisition, R.C.; Investigation, R.C., M.G., G.B. and A.G.; Methodology, R.C. and A.G.; Project Administration, R.C.; Supervision, R.C.; Visualization, R.C.; Writing—original draft, R.C.; Writing—review & editing, R.C. All authors have read and agreed to the published version of the manuscript.

**Funding:** The authors gratefully acknowledge financial support from Ministero dell'Istruzione, dell'Università e della Ricerca Italiana (MIUR).

**Institutional Review Board Statement:** The study was conducted according to the guidelines of the Declaration of Helsinki.

**Informed Consent Statement:** Informed consent was obtained from the subject involved in the study.

**Data Availability Statement:** Not applicable.

**Acknowledgments:** The authors thank D. Montesano for his technical assistance. Part of this work comes from the contribution by Giuseppe Pafetta's (2020) thesis and is fully acknowledge for its experimental skills.

**Conflicts of Interest:** The authors declare no conflict of interest.

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
