# Peer review of "Electrosynthesized Poly(o-aminophenol) Films as Biomimetic Coatings for Dopamine Detection on Pt Substrates"

_chemosensors, doi:10.3390/chemosensors9100280_

Round 1

Reviewer 1 Report

The paper by Ciriello et al. reports an eletrochemistry sensor for dopamine detection, using an MIP coated Pt eletrode. In my opnion, the major advance of this work is to employ PoAP as the target recognition unit. However, the advantage of PoAP should be highlighted and demonstrated, before further consideration of publication. Please find below some comments:

  1. It is still not clear what the advantage of PoAP is, compared with poly(pyrrole) used in Ref 20. The advantages should be demonstrated experimentally.
  2. The selectivity of this sensor should be demonstrated by testing AA, UA, SER,TYR before testing the mixtures.
  3. The abstract is too long. I suggest pointing out the novelty more straightforwardly.
  4. Pay attention to some errors in English, e.g. 'easy of operation'.

Reviewer 2 Report

Dear Authors,

I think that your work is well written and presented. Also, your scientific activity appears to be solid. I only have two marginal suggestions to give you:

-) Par. 2.2: please add the main parameters’ values for the DPV measurements.

-) Fig.2 and 4: please add colors or arrows to the various lines to indicate the progress of the experiments.

-) Fig. 6: in these experiments, the measured current is much smaller than that recorded in previous experiments, e.g., Fig. 5. Also, the values are now negative, and this could induce confusion in the casual reader. I suggest that you add few lines to explain these differences with respect to the previous presented data.

Kind regards.

Round 2

Reviewer 1 Report

The authors have addressed the questions and I suggest publication as it is.

Author Response

We thank the reviewer for suggesting the manuscript publication as it is. We have performed the minor spell check required.